



# Extending wind profile beyond the surface layer by combining physical and machine learning approaches

Boming Liu[1], Xin Ma[1], Jianping Guo[2*], Hui Li[1], Shikuan Jin[1], Yingying Ma[1], and Wei Gong[1]

[1]State Key Laboratory of Information Engineering in Surveying, Mapping and Remote Sensing (LIESMARS), Wuhan

University, Wuhan 430072, China

[2]State Key Laboratory of Severe Weather, Chinese Academy of Meteorological Sciences, Beijing 100081, China

*Correspondence to*: Dr./Prof. Jianping Guo (Email: jpguocams@gmail.com)

**Abstract:** Accurate estimation of the wind profile, especially in the lowest few hundred meters of the atmosphere, is of great significance for weather, climate and renewable energy. Nevertheless, the Monin–Obukhov similarity theory fails above the

surface layer over the heterogeneous underlying surface, resulting in an unreliable wind profile obtained from the conventional extrapolation methods. To solve this problem, we propose a novel method that combines the power law method (PLM) with the random forest (RF) algorithm to extend wind profiles beyond the surface layer, called the Phy-RF method. The underlying principle is to treat the wind profile as a power law distribution in the vertical direction, in which the power law exponent ($\alpha$) is determined by the Phy-RF model. First, the Phy-RF model is constructed based on the atmosphere sounding data at 119

radiosonde (RS) stations across China and in conjunction with other data such as surface wind speed, land cover type, surface roughness, friction velocity, geographical location, and meteorological parameters from June 2020 to May 2021. Afterwards, the performance of the Phy-RF, PLM and RF methods over China are evaluated by comparing them with RS observations. Overall, the wind speed at 100 m of the Phy-RF model exhibits high consistency with RS measurements, with a correlation coefficient (R) of 0.93 and a root mean squared error (RMSE) of 0.92 m s$^{-1}$. By contrast, the R and RMSE of wind speed

results from PLM (RF) method are 0.87 (0.91) and 1.37 (1.04) m s$^{-1}$, respectively. This indicates that the estimates from the Phy-RF method are much closer to observations than those from the PLM and RF methods. Moreover, the RMSE of the wind profiles estimated by the Phy-RF model is relatively larger at highlands, while is small in plains. The result indicates that the performance of the Phy-RF model is affected by the terrain factor. Finally, the Phy-RF model is applied to three atmospheric radiation measurement sites for independent validation, and the wind profiles estimated by the Phy-RF model are found

consistent with the Doppler Lidar observations. This confirms that the Phy-RF model has good applicability. These findings have great implications for the weather, climate and renewable energy.

## 1. Introduction

The atmospheric wind field is a critical factor in the transportation of water vapor and matter, influencing weather forecasting and climate change (Stoffelen et al., 2005; 2006). The wind profile, is a key parameter for measuring atmospheric wind field,

including turbulent mixing, convective transport, and material diffusion in the atmosphere (Solanki et al., 2022; Stoffelen et al., 2020). Particularly in the lowest few hundred meters of the atmosphere, the wind profile plays a significant role in wind energy assessment and in understanding the interactions between the atmosphere and the land (Gryning et al., 2007; Veers et al., 2019). Therefore, it is crucial to accurately comprehend the spatial distribution and dynamic variation of wind profiles.

Currently, there are multiple methods for observing wind profiles. Atmospheric reanalysis data, such as the fifth generation

ECMWF reanalysis (ERA5) which is based on known physical mechanisms combined with the assimilation of vast amount of observational data, has been widely used to derive the spatiotemporal distribution of wind profiles (Laurila et al., 2021; Gualtieri, 2021). Satellite observations, like Aeolus, can provide the horizontal line of sight wind profile observations, in which can be assimilate into atmospheric models to produce global wind profile products (Stoffelen et al., 2006; Guo et al., 2021). Nevertheless, the accuracy of wind profile products from Aeolus and ERA5 within the planetary boundary layer (PBL) requires



improvement due to factors such as atmospheric attenuation and turbulence (Straume et al., 2020; Deng et al., 2022). On the other hand, ground-based observations like wind tower, wind profile radar, and wind profile lidar, can obtain highly precise wind profile in the PBL at the observation station (Durisic et al., 2012; Wu et al., 2022). However, single site observations cannot acquire wind profile data on a regional or national scale. Therefore, researchers are endeavoring to develop a theoretical model for wind profile to acquire the large-scale PBL wind profiles.

The wind profile model was initially developed using the famous Monin–Obukhov similarity theory, which describes the wind profile as functions relying on a stability parameter h/L (Obukhov, 1946; Monin and Obukhov, 1954). The h stands for height while L stands for Obukhov length on the surface. The wind profile model based on similarity theory can be expressed in different forms depending on the varying atmospheric conditions. For the neutral conditions, the wind speed profile model can be simplified as a logarithmic law (Powell et al., 2003; Marusic et al., 2013). For the unstable conditions, the exponential wind

speed profile can better describe the wind speed profile in surface layer over homogeneous terrain (Barthelmie et al., 2020). In engineering applications, most studies utilize a power law model to consider the wind profile in the surface layer (Sen et al., 2012; Jung et al., 2021). This can achieve the conversion of surface wind speed to the wind speed at wind turbine hub height. These wind profile models based on the Monin–Obukhov similarity theory have demonstrated effectiveness within the surface layer (approximately 100 m above the ground). Nevertheless, due to factors such as Coriolis parameter, baroclinity, and wind

shear, the applicability of the Monin–Obukhov similarity theory breaks down above the surface layer (Optis et al., 2016; Tong et al., 2020). Therefore, extending wind profiles above the surface layer is of significance in applying wind profiles to wind energy assessment and PBL dynamics.

Above the surface layer, the wind profiles are influenced not only by the surface roughness, friction velocity and the atmospheric stability, but also by factors including low-level jets, entrainment processes, and the Coriolis parameter (Gryning

et al. 2007; Coleman et al., 2021). To obtain accurate wind profiles above the surface layer, some studies seek to introduce auxiliary variables to account for the influence of these factors. Gryning et al (2007) established a straightforward model that regulates the combined length scale of wind profiles along with their stability correlations. This model is used to calculate wind profiles above the surface. On the other hand, Liu et al. (2022) present an analytical approach based on the Ekman equations and the foundation of the universal potential temperature flux profile. This approach enables one to describe the

profiles of wind and turbulent shear stress, which in turn can capture aspects such as the wind veer profile. In addition, some studies use the machine learning (ML) technology to transform surface wind speed and meteorological parameters to wind speeds at different heights. Yu et al. (2022) has devised a transfer method that leverages three ML methods, including the least absolute shrinkage selector operator, random forest (RF) and extreme gradient boost for calculating wind speed at 100 m. Liu et al. (2023) employed the RF model to estimate the wind speed at 120 m, 160 m and 200 m. Nevertheless, the calculation

procedure of ML algorithms remains an unexplained process that does not clarify the input parameter's physical significance. Therefore, it is worth trying to combine ML algorithms with physical models to achieve the inversion of wind profiles above the surface layer.

The present study aims to extend wind profiles beyond the surface layer by combining the physics and machine learning approaches. For this purpose, we attempt to combine the power law method (PLM) with the RF, named Phy-RF model, to

extend wind profiles beyond the surface layer. The Phy-RF model is trained and tested using radiosonde (RS) data and reanalysis gridded meteorological data over China. The performance of the PLM, RF and Phy-RF models is also compared. Then, the wind profile generated by the Phy-RF model is evaluated against RS observations, followed by independent validation of the model at atmospheric radiation measurement (ARM) sites. The results of our study have great implications for the weather, climate and renewable energy.



## 2. Materials and Data

### 2.1 Land cover type data

The land cover type data is derived using the Moderate Resolution Imaging Spectroradiometer (MODIS), a satellite-borne instrument that captures images and measures diverse a wide range of surface properties such as land surface temperature, vegetation cover, and atmospheric aerosols (Friedl et al., 2002). The high spatial resolution of the instrument enables the identification of diverse land features, including forests, urban areas, and agricultural fields, thereby making it an important instrument for the purpose of environmental monitoring and land management (Sulla-Menashe et al., 2018). The MODIS provides two land cover type products: MCD12Q1 and MCD12C1. While MCD12Q1 comprises observation data from different regions, which requires self-splicing, MCD12C1 is an annual concatenated data containing one image per year. Following the previous study (Liu et al., 2020), the land cover type data here is obtained from the MCD12C1, named "MCD12C1.A2021001.061.2022217040006". Fig. 1 displays the dominant land cover types' geographic distribution in China. The colorbars from bottom to top represent Water, Evergreen Needleleaf Forest, Evergreen Broadleaf Forest, Deciduous Needleleaf forest, Deciduous Broadleaf Forest, Mixed forest, Closed shrublands, Open shrublands, Woody savannas, Savannas, Grasslands, Permanent, Croplands, Urban and built-up, Cropland/Natural vegetation mosaic, Snow and ice, and Barren or sparsely vegetated, respectively. These MODIS products help us to determine the power law exponent ($\alpha$).

### 2.2 Radiosonde measurements

The L-band RS can measure the profiles of atmospheric temperature, pressure, humidity, wind direction and wind speed in-situ. It is taken at 1-min intervals starting from the ground surface up to approximately 30 km above ground level (Guo et al., 2016). The RS observations are conducted at 119 observation stations in China, which is shown in Fig.1. The RS are launched twice per day at around 0800 and 2000 local time (LT). Here, the wind speed profiles from RS measurement at 119 stations are obtained as reference value (National Meteorological Science Data Center, 2023). The RS collection time is from 1 June 2020 to 30 May 2021. In addition, the drift of the RS during its ascent has been investigated, as illustrated in Fig. S1. The RS and observation stations' coordinates drifting distance within a height of 0.5 km is less than 0.5 km. This indicates that the drift of RS will not impact obtaining wind profiles in surface layer.

### 2.3 ERA5 data

The ERA5 is a fifth-generation reanalysis dataset that offers a range of atmospheric parameters, such as temperature, humidity, pressure and radiation (Hersbach et al., 2020). Following previous study (Liu et al., 2023), nine surface parameters have been obtained in this study, including charnock coefficient (Char), forecast surface roughness (FSR), friction velocity (FV), dew point (DP), temperature (Temp), pressure (Pres), net solar radiation (Rn), latent heat flux (LHF), and sensible heat flux (SHF). These parameters are processed into grid data with a 0.25 × 0.25 size and an hourly time resolution. Based on the longitude and latitude information of the RS and ARM stations, those parameters in the corresponding grid are obtained accordingly. These data are also collected for the period spanning from 1 June 2020 to 30 May 2021.

### 2.4 ARM data

The ARM user facility is established by the U.S. Department of Energy (Lubin et al., 2020; Zhang et al., 2022). It sets up observation stations and instruments globally for atmospheric observation experiments, making publicly assessable the atmospheric observations, including temperature, wind, radiation, and cloud properties (Liu et al., 2022). The wind profile data from the Doppler Lidars deployed at the Eastern North Atlantic (ENA), North Slope of Alaska (NSA), and Southern Great Plains (SGP) stations are collected to independently compare the proposed method. Figure S2 presents the geographic locations and land cover types of the three stations. It can be seen that ENA is situated on an Atlantic Ocean Island ocean as its primary land cover, NSA is situated on Alaska's north coast with grassland as its ground cover, and SGP is located in the Great Plains



in the central United States where grassland is also the dominant land cover. The Doppler Lidar observations covers the period
from 1 June 2020 to 30 May 2021. Moreover, these wind profiling measurements are processed as hourly averages correspond
with other data.

### 3. Methods

#### 3.1 Power law method

The PLM assumes that wind speed increases exponentially with height (Hellman et al. 1914). The wind profile can be
calculated based on the surface wind speed ($v_0$) using the following formula:

$$v_i = v_0 \times \left(\frac{h_i}{h_0}\right)^{\alpha} \tag{1}$$

where $v_i$ represents wind speed at height $h_i$. The $h_0$ is the measurement height of $v_0$. Here, the $v_0$ is observed by an anemometer
at a height of 10 m above the ground. The $\alpha$ is the power law exponent, which varies with land cover type, height and time (Li
et al., 2018).

The $\alpha$ is usually set as a constant 0.14 for the purpose of approximating the wind profile at station where there are no
observations or empirical formulas available. Figs. 2a and 2c show the RMSE and difference between PLM ($\alpha$=0.14) results
and RS measurements for wind speed at 100 m (WS$_{100}$). The average RMSE and difference over China are 1.49±0.39 and -
0.23±0.68 m s$^{-1}$, respectively. The results indicate that PLM results ($\alpha$=0.14) underestimate wind profiles at almost a quarter
of the sites (Fig. 2c). These results suggest that the estimation of wind profiles based on a constant $\alpha$ value is subject to large
errors. Some studies also confirm it (Jung et al., 2021; Liu et al., 2023). Furthermore, other studies demonstrate that the $\alpha$
values differ based on the land cover type due to varying surface roughness (Durisic et al., 2012). An empirical lookup table
is summarized with respect to the setting of $\alpha$, as shown in Table S1. The value of $\alpha$ ranges from 0.1 to 0.4 with increasing
surface roughness. Based on the MODIS land cover type dataset, the corresponding value of $\alpha$ can be obtained for all RS sites.
Figs. 2b and 2d show the RMSE and the difference between the PLM (dynamic $\alpha$) results and RS measurements. Compared
with the PLM ($\alpha$=0.14), the results of PLM (dynamic $\alpha$) have improved. However, the results of PLM (dynamic $\alpha$) are still
underestimated at most stations in the Northeast and Inner Mongolia regions.

#### 3.2 Random forest model

RF model is a nonlinear fitting algorithm and has been used to calculate wind profiles (Yu et al., 2022; Liu et al., 2023). Here,
the RF model is also used to fit the surface parameters to obtain the wind profile. The input variables include surface wind
speed (WS), surface wind direction (WD), land cover type (Type), altitude (Alt), longitude (Lon), latitude (Lat), month (M),
hour (H), Char, FSR, FV, DP, Temp, Pres, Rn, LHF, and SHF. The reference value is the wind speed provided by RS. In
addition, the parameter tuning of the RF model directly affects the performance and generalization ability of the model (Zhu
et al., 2021). The tuning parameter process for the Estimator number and Min Leafsize is shown in Fig. S3. The RMSE is
minimum (1.02 m s$^{-1}$) and R is maximum (0.91) when the Estimator number is 300 and the Min Leafsize is 5. Therefore, the
Estimator number and the Min Leafsize is set to 300 and 5 for the RF model, respectively.

#### 3.3 Combining physical and RF model

In this study, we propose a novel method, termed Phy-RF that combines physical and RF model, to estimate wind profiles. Its
principle is to treat the wind profile as a power law distribution in the vertical direction where the $\alpha$ is fitted by using the RF
model. The details for this method go as follows:

#### 3.3.1 Physical constraint



Previous studies have confirmed that the wind profile in the surface layer adheres to a power law distribution. The primary reason for the error is the uncertainty in the value of $\alpha$. Therefore, to achieve more accurate results, it is necessary to first analyze the reasons of the error. Figs.3a and 3b display the differences in $WS_{100}$ estimated by PLM ($\alpha$=0.14) and PLM (dynamic
$\alpha$) relative to RS observations. The color bar indicates the various land cover types. Based on MODIS land cover type data, the 119 RS sites are classified as urban area, woodland, shrubs, grassland, and smooth surface. It is found that regardless of land cover type, the difference in wind speed decreases as the surface wind speed increases. Similarly, the differences between assumed $\alpha$ and observed $\alpha$ at 100 m decrease with increasing surface wind speed (Figs. 3c and 3d). These results indicate that there is a relationship between the error of the PLM results and the surface wind speed. This may be due to the limited influence
of surface friction on the wind profile. When the wind speed within the PBL is low, factors such as surface friction and Coriolis force complicate the vertical distribution of the wind profiles, leading to the low surface wind speed and large errors in the PLM (Wang et al., 2023). On the other hand, when the wind speed within the PBL is high, the effect of surface friction can be neglected to some extent. This results in the real wind profile being closer to the power law distribution, thereby reducing the error of the PLM results.

To quantify the effect of surface wind speed on $\alpha$, the $\alpha$ bias (assumed value minus observed value) at 100 m is examined as a function of surface wind speed under different land cover types is investigated, as shown in Fig.4. The gray dots and black lines indicate the sample points and the logarithmic curve, respectively. The coefficients of determination between the surface wind speed and the difference of $\alpha$ on all types, urban area, woodland, shrubs, grassland, and smooth surface are 0.92, 0.97, 0.94, 0.97, 0.93 and 0.84, respectively. It indicates that there is a good correlation between the surface wind speed and
difference of $\alpha$. Therefore, the correction factor of $\alpha$ ($\Delta\alpha$) can be defined statistically based on the land cover type and surface wind speed. The correction functions of $\alpha$ for different land cover types are also plotted in Fig. 4. For each sample, the $\Delta\alpha$ can be calculated by the correction functions, and then entered the fitting of the RF model as a physical constraint to improve the accuracy. In addition, the $\alpha$ bias as a function of surface wind speed at different heights is also investigated, as shown in Fig. S4. At 50, 100, 150, 200, 250 and 300 m, the coefficients of determination between the surface wind speed and the difference
of $\alpha$ are larger than 0.9. This indicates that the $\Delta\alpha$ can be constructed by the surface wind speed to improve the inversion accuracy of the wind speed at high-altitude.

### 3.3.2 Model construction

For the Phy-RF model, the wind profile is considered as a power law distribution, and the $\alpha$ is fitted by the RF model. The inputs include $\Delta\alpha$, WS, WD, Type, Alt, Lon, Lat, M, H, Char, FSR, FV, DP, Temp, Pres, Rn, LHF, and SHF. The reference
value is the $\alpha$ calculated from RS observations. The tuning parameter evolution for the Phy-RF model is shown in Fig. S5. The RMSE reaches a minimum (0.91 m s$^{-1}$) and R reaches a maximum (0.93) when the Estimator number is 500 and Min Leafsize is 5. Therefore, the Estimator number and Min Leafsize are set to 500 and 5, respectively.

To comprehend the model's physical meaning, the importance analysis of the inputs is performed for the RF and Phy-RF models, as shown in Fig. 5. The relevant features that can affect the accuracy of the model accuracy are marked with red bars.
For the RF model, the relevant features are WS, Type, SHF, FV, WD and FSR. The importance of WS, Type and SHF is greater than other features. WS is the surface wind speed. Type is the value of $\alpha$ based on the land cover type. From the perspective of physical meaning, the RF model calculates wind profiles through complex fitting methods based on surface wind speed and meteorological conditions. In contrast, for the Phy-RF model, $\Delta\alpha$, FV, SHF, Type, WS, FSR and Temp are both the relevant features. The importance of $\Delta\alpha$ is the largest, but the importance of Type and WS are ranked fourth and fifth.
In addition, the importance of FV ranks second. FV is used to calculate the way the wind changes with height in the lowest levels of the atmosphere (Liu et al., 2023). These results indicate that the Phy-RF model calculates the way wind speed changes in the vertical direction. In addition, the SHF and FSR are both relevant features in the construction of the RF and Phy-RF



models. This indicates that surface roughness and solar radiation are factors that need to consider in the calculation of wind profiles.

**3.4 Sensitivity analysis**

The average values and standard deviations of the variance between the assumed $\alpha$ and the observed $\alpha$ are illustrated for the primary input features (Fig. 6). Green, blue and red represent the PLM, RF, and Phy-RF models, respectively. The differences in deviations for Phy-RF models decrease slightly with increasing surface wind speed. Moreover, the means and deviations of differences for the Phy-RF model are relatively stable and do not vary with the land cover types. These results indicate that both the RF and Phy-RF models exhibit good generalization across different land cover types and surface wind speeds. This is due to that fact that the RF model considers random perturbations in the sample space to improve generalization ability (Breiman, 2001). In addition, due to the samples are only obtained at 08:00 and 20:00 LT, it notes that whether the performance of the Phy-RF model is affected by time. The RS observation stations are geographically distributed in several time zones, but they are all observed at the same time. This means that although the recording time of the RS measurements is 08:00 and 20:00 LT, the training and test samples contain observations data from multiple time periods. Therefore, Rn is used as a measure of time to investigate the applicability of the methods (Fig. 6c). For three methods, the means of the difference are relatively stable and the standard deviations of the difference decrease slightly as Rn increases. This indicates that the generalization of the Phy-RF model within the sample is reliable. However, the Rn in China at noon can reach $1.5 \sim 2*10^6$ J/m$^2$, which exceeds the upper limit of the input values in the current sample. It indicates that the generalization of the Phy-RF model at noon time cannot be proven based on the existing training and test samples. Therefore, it can only rely on the LiDAR data from the ARM sites for comparison to evaluate the performance of the Phy-RF model at noon. Specific comparisons will be discussed in Sect. 4.4.

**4. Results and discussion**

In this section, the performances of PLM, RF and Phy-RF models are firstly compared by conducting intercomparison analyses. The wind profiles calculated by the Phy-RF model are then evaluated by comparing with the RS observations. Finally, the Phy-RF model is applied to three ARM sites for independent validation.

**4.1 Intercomparison of different methods.**

Fig. 7 displays the scatter plot between the estimated $WS_{100}$ and the observed $WS_{100}$ for three methods at different times. Overall, the R (RMSE) of $WS_{100}$ from PLM, RF and Phy-RF under all times are 0.87 (1.37 m s$^{-1}$), 0.91 (1.04 m s$^{-1}$) and 0.93 (0.92 m s$^{-1}$), respectively. The accuracy of the RF and Phy-RF models is better than that of the PLM. For the PLM, most of the estimated $WS_{100}$ are underestimated when the observed value is high. This is because the PLM relies on the exponential relationship to calculate the $WS_{100}$. However, the wind profile is affected by turbulence, surface friction and other factors (Tieleman 1992; Solanki et al., 2022). The exponential law based on constants is unable to obtain the $WS_{100}$ with high accuracy. In contrast, the performance of the RF and Phy-RF models improved significantly. The RF and Phy-RF models consider more environmental factors, such as SHF and FV, in the inversion process. They improve the accuracy of the model by considering the effects of surface friction and surface radiation flux on the wind profiles. Briefly, these two methods rely on a dynamic $\alpha$ to invert the wind profiles. Each site uses the $\alpha$ that varies with environmental factors, resulting in improved inversion accuracy. In particular, for the Phy-RF model, the $\alpha$ correction function can be used to obtain a value of $\alpha$ closer to the observation, resulting in the highest R (0.94) and the lowest RMSE (1.00 m s$^{-1}$). In addition, the RMSE of $WS_{100}$ from PLM, RF and Phy-RF at 08:00 (20:00) LT are 1.39 (1.35), 1.05 (1.03) and 0.94 (0.91) m s$^{-1}$, respectively. The comparison results at 08:00 and 20:00 LT are also show that the performance of Phy-RF is the best, followed by RF; last are PLM.





Figure 8 shows the difference and RMSE between the estimated $WS_{100}$ and the observed $WS_{100}$ for three methods under different months. Overall, the mean difference of the $WS_{100}$ from three methods is close to 0, but the quantile range of the difference is not consistent. For PLM, the quantile ranges of the difference are wider during the cold season (October–April) than during the warm season (June–September). This is because the wind speed variations are more complex during the cold season. The large-scale synoptic systems have a relatively high frequency of occurrence during the cold season (Liu et al., 2019). Compared with PLM, the RF and Phy-RF models have the stable accuracy over the 12 months, i.e., the difference between the months is relatively small. In addition, the quantile range of the Phy-RF model is narrower than that of the other models. The average difference for the $WS_{100}$ of Phy-RF model does not show significant seasonal differences. It indicates that the Phy-RF model is not affected by seasonal variation. This is because that the RF models are data-driven (Zhu et al., 2021; Ma et al., 2021). After correcting the $\alpha$ based on the RF model, the Phy-RF model can effectively overcome the influence of seasonal factors. Fig. 8b shows that the $WS_{100}$ from Phy-RF model has a smaller RMSE among the months, and the RMSE is relatively stable over the 12 months. The results indicated that the Phy-RF model outperforms both PLM and RF in terms of accuracy and stability. Therefore, the Phy-RF model may be a more suitable choice for estimating wind profiles in China than either RF or PLM.

**4.2 Evaluation the wind speed of Phy-RF model**

Fig. 9 shows the spatial distributions of the mean wind speed from ERA5 (color shaded) and Phy-RF model (colored dots) at 100 m for different periods. In general, the mean wind speeds of ERA5 and Phy-RF model are generally similar. About the seasonal variation, the $WS_{100}$ is low in summer and fall and high in spring and winter. This is due to the frequently large-scale synoptic systems in the cold season (Liu et al., 2019). From the perspective of spatial distribution, the $WS_{100}$ is the highest in Inner Mongolia and Northeast China, followed by coastal areas, and the lowest in inland areas. There are two reasons for the high wind speed in Inner Mongolia and Northeast China. One is that the climate in these areas is dry and cold, especially in winter. The low temperature and high air density lead to the formation of a strong pressure gradient (Liu et al., 2019). When the pressure gradient is large, cyclonic and anticyclonic weather will occur, resulting in higher wind speed. Another reason is that these areas are susceptible to the influence of the Siberian monsoon and warm currents from the Pacific (Yu et al., 2016). This monsoon will cause an increase in wind speed as it passes through Inner Mongolia and Northeast China. In addition, the spatial distributions of wind speed estimated by the Phy-RF model at different heights are shown in Fig. S6. In the vertical direction, the wind speed gradually increases with height. Especially at 200 to 300 m, the wind speed is significantly higher than the $WS_{100}$. This is because the surface wind profile is affected by the friction velocity and surface roughness (Gryning et al., 2007). The obstructions such as trees, buildings, and hills cause turbulence and reduce the wind speed (Solanki et al., 2022).

Figure 10 shows the spatial distributions of the difference between the estimated wind speed and the RS observation for the Phy-RF model at different heights. At 50 and 100 m, most sites (more than 90%) show a mean difference of less than 0.2 m s$^{-1}$, with an overall mean difference of -0.02±0.02 m s$^{-1}$ and -0.15±0.05 m s$^{-1}$, respectively. In contrast, above 100m, the average differences are negative at almost all sites. The mean differences for all sites at 150, 200, 250 and 300 m are -0.19±0.08, -0.24±0.10, -0.24±0.10, and -0.25±0.11 m s$^{-1}$, respectively. Compared to the results of PLM (Fig. 2c and 2d), the accuracy of the wind speed in the Phy-RF model has improved. Overall, the wind speed estimated by the Phy-RF model is slightly underestimated compared to the observed values. Moreover, the average difference gradually increases with the increasing height. This is because the wind profile above the surface layer is not logarithmic but increases faster in response to the reduction in surface friction force (Gryning et al., 2007; Liu et al., 2023). The mean RMSE between the estimated and observed wind speeds at different heights also confirms that the performance of the Phy-RF model decreases with increasing height (Fig. S7). This is because the wind profile above the surface layer is affected by the influence of the low-level jets, entrainment processes, and the Coriolis parameter (Coleman et al., 2021). The introduction of $\Delta\alpha$ aimed to enhance the performance of the Phy-RF model, but the results indicate that the Phy-RF model slightly underestimates at high altitudes.



### 4.3 Effect of terrain

To evaluate the effect of terrain factor on the performance of Phy-RF model, the plain terrain is defined, which the topographic relief is less than 50 m within a radius of 5 km around the observation station. The RS sites are divided into two categories: plains (marked by red dots) and highlands (marked by black dots), as show in Fig. S8. Fig. 11a shows the mean $\alpha$ observed from RS at different height. Blue and red boxes represent the results under plains and highland areas, respectively. The mean $\alpha$ in highlands is greater than that in plains. It indicates that the variation of wind profiles in highlands is more complex than

that in plains. Previous studies have also shown that the valley winds and low-level jets can complicate the wind profiles in the PBL (Solanki et al., 2021; Wang et al., 2023). Fig 11b shows the difference between the estimated and observed wind speed for Phy-RF model at different height. The difference in highlands is obviously larger than that in plains. Moreover, the similar phenomena were also found in the results of RMSE (Fig. 11c). The RMSE in highlands is relatively large, while is relatively small in plains. This may be due to differences in terrain. The terrain in plains is mainly flat, while the terrain in

highlands is mainly mountainous (Chen et al., 2016). The wind profile is not only affected by factors such as surface friction and solar radiation, but also constrained by the terrain (Panofsky et al., 1964; Jung et al., 2021). In the construction of the Phy-RF model, the influence of the terrain factor was not considered, resulting in a higher RMSE of the Phy-RF model on the highlands.

### 4.4 Independent validation

Figure 12 displays the vertical wind speed distribution using different methods at three ARM sites. At the NSA site, the wind profiles calculated by the Phy-RF model are similar to the observed values at 08:00 and 20:00 LT, but are overestimated at 14:00 LT. The performance of the Phy-RF model at 14:00 LT is inferior to that of PLM (Fig. 12c). Similarly, this phenomenon occurs at sites of SGP as well. The results of the Phy-RF model are significantly overestimated at 14:00 LT. These results indicate that the performance of the Phy-RF model is influenced by hourly variations. The generalization of the RF algorithm

depends on the training and test samples (Zhu et al., 2021). As mentioned in Sect. 3.4, the training and test samples of the Phy-RF model do not actually contain data from 11:00 to 15:00 LT time periods. This means that the Phy-RF model has no generalization at noon, resulting in poor accuracy of the Phy-RF model at 14:00 LT. However, to our surprise, the result of the Phy-RF model is very consistent with the observations at the ENA site, even at 14:00 LT (Fig. 12g). This may be due to the differences in land cover types between the sites. NSA and SGP sites are located on land, with significant diurnal variations

in wind speed. The wind speed at daytime is relatively low, even lower than the estimated value of PLM. In contrast, the ENA site is located on the island, and the diurnal variation of wind speed is not significant. The wind speed throughout the day is higher than the estimated value of PLM. For the Phy RF model, since the training data is mainly composed of relatively high wind speed at nighttime, the model exhibits significant overestimation correction. The model can accurately calculate wind speed when the actual value is larger than the estimated value of PLM, while it will significantly overestimate the actual value

if it is lower than the estimated PLM. Although the Phy-RF model has some overestimation at 14:00 LT, the comparisons at other times indicate that the wind profiles of the Phy-RF model are still similar to the observed results (Fig. 12a, 12e and 12i). Overall, the Phy-RF model's wind profiles exhibit greater proximity to the observed values when compared to the results generated by PLM at three ARM sites. Additionally, the wind speed results retrieved by the Phy-RF model are consistent with Lidar's observations at different heights. These results indicate that the Phy-RF model has good spatial applicability and can

be used to obtain the wind profiles on different land cover types.

### 5. Summary and conclusions

The traditional wind profile model was constructed based on the Monin–Obukhov similarity theory. As a result, the wind profile based on the similarity theory is only effective within the surface layer. To address this challenge, this study proposes



a Phy-RF method that combines the traditional PLM with the RF algorithm to extend the wind profiles beyond the surface
layer.

The reasons for the errors in PLM above the surface layer are first analyzed. The result indicates that the error of PLM is mainly attributable to the $\alpha$ setting. This is because the wind profile above the surface is affected by factors such as surface roughness, friction velocity, low-level jets, and Coriolis parameter, resulting in the complexity of the $\alpha$. Moreover, the surface wind speed has a certain impact on the variation of $\alpha$. At the height of 50, 100, 150, 200, 250 and 300 m, the coefficients of
determination between the surface wind speed and the difference of $\alpha$ are greater than 0.9. It may be due to the limited influence of surface friction on the wind profile. When the PBL wind is high, the effect of surface friction can be neglected to some extent, resulting in the real wind profile being closer to the power law distribution. Based on this physical constraint, the Phy-RF method considers the wind profile as a power law distribution in the vertical direction, and the $\alpha$ at different heights is fitted by the RF model to calculate the wind profile.

The performance of PLM, RF and Phy-RF methods are compared based on the RS observations over China from 1 June 2020 to 30 May 2021. The R (RMSE) of the $WS_{100}$ from PLM, RF and Phy-RF models were 0.87 (1.37 m s−1), 0.91 (1.04 m s−1) and 0.93 (0.92 m s−1), respectively. It shows that the Phy-RF model has better accuracy and stability compared to PLM and RF. The Phy-RF model can be understood as the PLM based on dynamic $\alpha$. The RF model is used to adjust $\alpha$ at different heights based on factors such as surface wind speed, land cover type, and meteorological parameters to achieve high-precision
wind profile inversion. Afterwards, the wind profiles calculated by the Phy-RF model are then evaluated from a temporal and spatial perspective. From a temporal distribution perspective, the average difference for the Phy-RF model does not exhibit significant seasonal variations. It indicates that the Phy-RF model is not affected by seasonal variation. This is because the RF model is data driven. The training sample of the Phy-RF model contains enough samples from four seasons. In contrast, the Phy-RF model is affected by diurnal variation. The generalization of the RF model depends on whether the training samples contain sufficient sample inputs. The training samples of the Phy-RF model do not contain data from 11:00 to 15:00 LT time
periods, resulting in poor accuracy during this period. Therefore, it is not recommended to use the Phy-RF model during 11:00 to 15:00 LT before adding observation data to retrain the model. From a spatial distribution perspective, the RMSE of the wind profiles is relatively larger at highland areas. It indicates that the Phy-RF model is affected by the terrain factor. This is due to the influence of terrain factor was not considered in the construction of the Phy-RF model. Overall, the Phy-RF model can
provide more accurate wind profiles than the PLM and RF models.

Our study extends the wind profile beyond the surface layer by combining physical and ML approaches, which has great implications for the weather, climate and renewable energy. However, due to the limitations of data time and terrain factors, it should pay attention to the application scenarios of the Phy-RF model. In the future, the topographic conditions need more attention and deserve further investigation.

**Data Availability**

The output data and codes used in this paper can be provided for non-commercial research purposes upon reasonable request (Jianping Guo, email: jpguocams@gmail.com). The RS data can be downloaded from http://www.nmic.cn/data/cdcdetail/dataCode/B.0011.0001C.html (National Meteorological Science Data Center, 2023). The ERA5 data can be downloaded from https://cds.climate.copernicus.eu/cdsapp#!/dataset/reanalysis-era5-single-
levels?tab=overview (ECMWF, 2023). The ARM data can be downloaded from https://adc.arm.gov/discovery/#/results/instrument_class_code::dlprof-wind (Atmospheric Radiation Measurement (ARM) user facility data, 2023).



**Acknowledgments**

This work was supported by the National Natural Science Foundation of China (under grants 42001291 and 42325501), the Fundamental Research Funds for the Central Universities (under grants 2042022kf1003), and the Natural Science Fund of Hubei Province (under grants 2022CFB044).

**Author Contributions**

The study was completed with cooperation between all authors. JG and BL designed the research framework; BL and JG conducted the experiment and wrote the paper; XM, HL, SJ, YM, and WG analyzed the experimental results and helped touch on the manuscript.

**Competing interests**

The authors declare that they have no conflict of interest.

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



**Figures:**

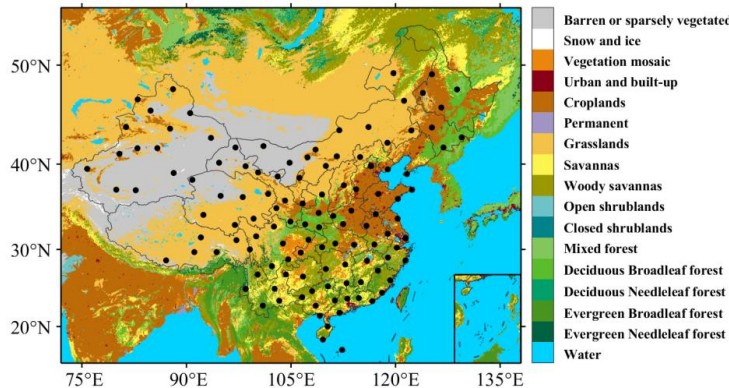


**Figure 1.** Geographical distribution of the radiosonde stations in China, which is overlaid over surface land cover type (color shading) from the MODIS observations.



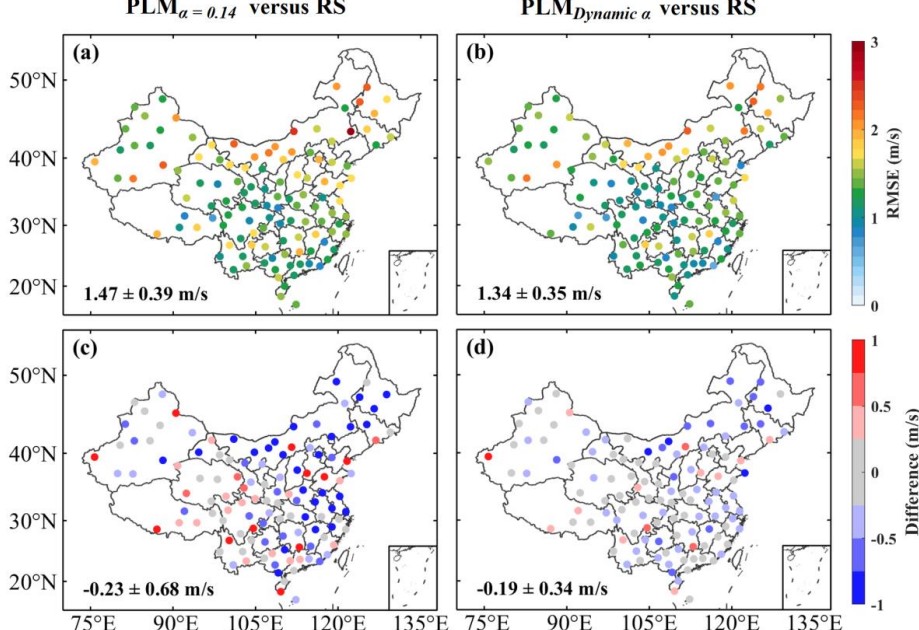


**Figure. 2** The spatial distributions of (a, b) RMSE and (c, d) difference for the WS$_{100}$ estimated by the traditional PLM (constant $\alpha$ of 0.14) and PLM (dynamic $\alpha$), respectively. Also shown in the bottom left corner are the mean values of RMSE and difference.



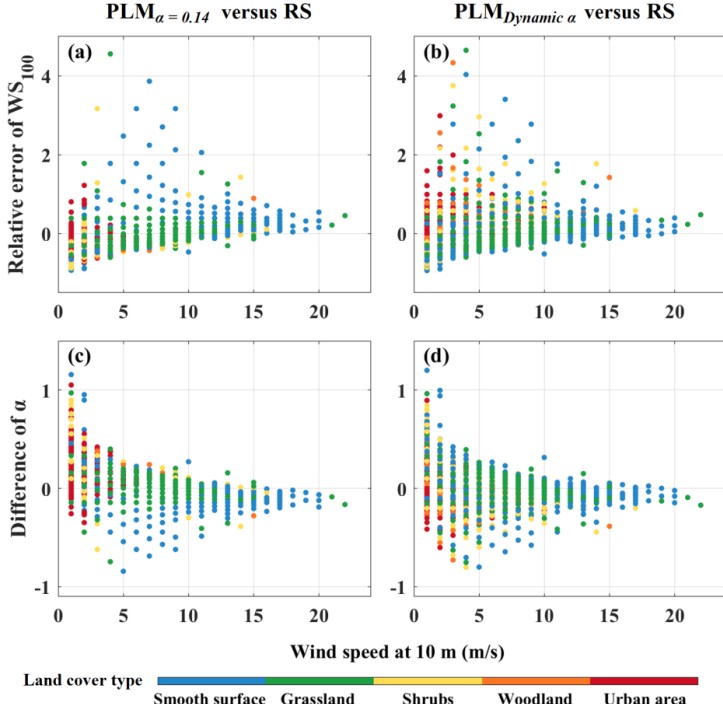

**Figure 3.** The differences in (a, b) $WS_{100}$ and (c, d) $\alpha$ estimated by PLM ($\alpha$=0.14) and PLM (dynamic $\alpha$) relative to RS observations shown as a function of land cover type (color shading).







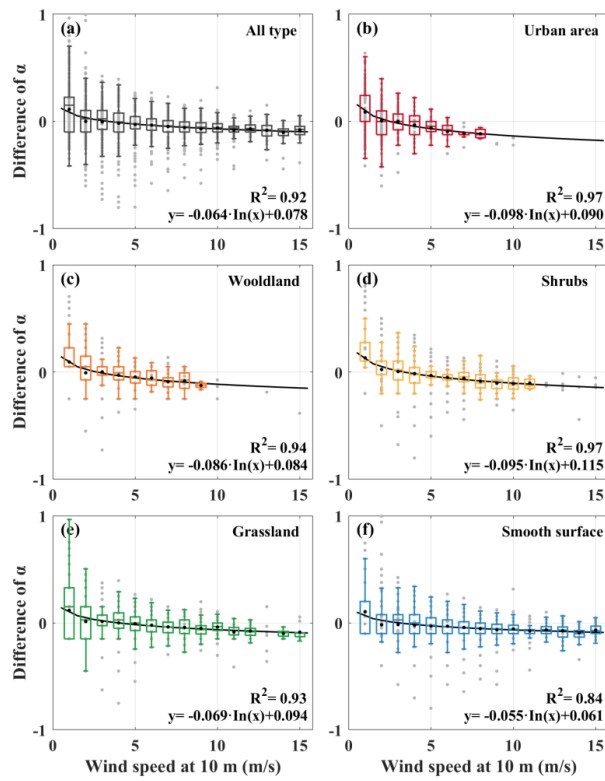

**Figure 4**. The $\alpha$ bias (assumed value minus observed value) at 100 m as a function of surface wind speed under (a) all types of land cover, (b) urban area, (c) wooldland, (d) shrubs, (e) grassland, and (f) smooth surface.



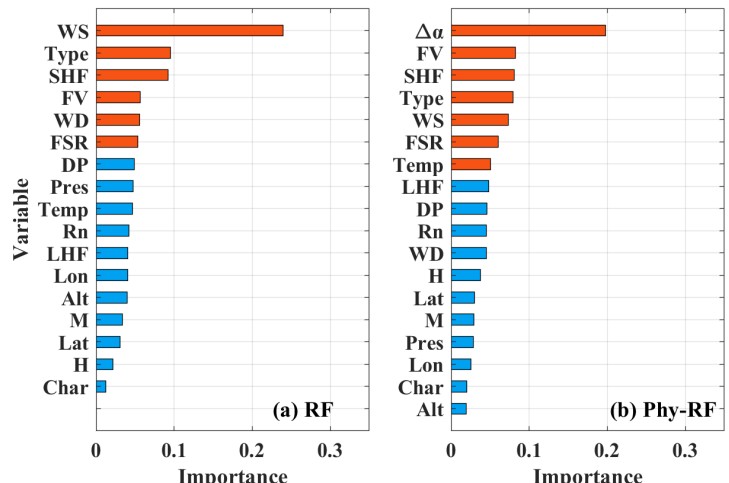

**Figure 5.** The importance analysis of inputs for the (a) RF and (b) Phy-RF models.




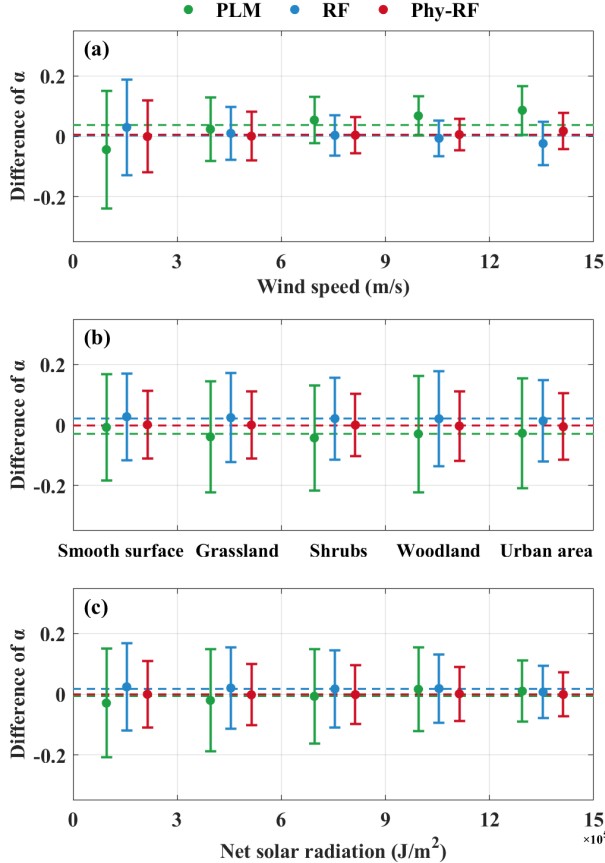

**Figure 6.** The means and standard deviations of the difference between assumed $\alpha$ and observed $\alpha$ at 100 m as a function of
(a) surface wind speeds, (b) land cover type and (c) net solar radiation. Green, blue, and red lines represent the results from
the PLM, the RF, and the Phy-RF, respectively.



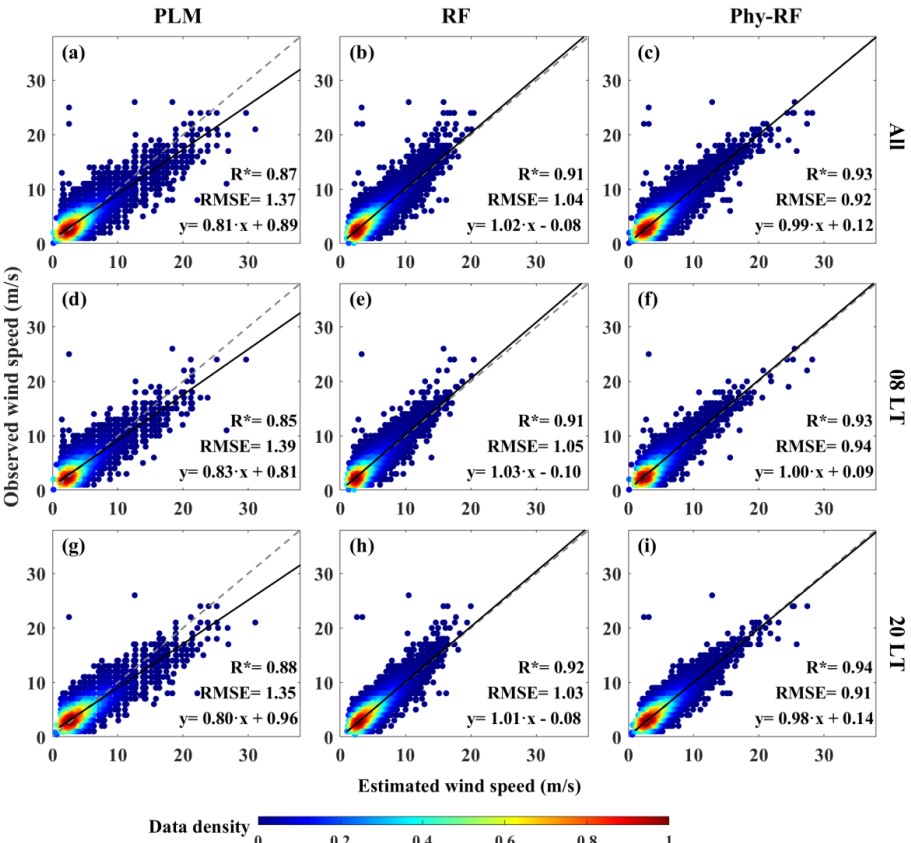

**Figure 7.** Comparisons between observed $WS_{100}$ and estimated $WS_{100}$ for (a, d, g) the PLM, (b, e, h) RF and (c, f, i) Phy-RF models under all time, 0800 LT and 2000 LT. The gray and black lines are the reference and regression lines, respectively. The color bar represents the data density. The asterisk indicates that the correlation coefficient (R) has passed the t test at a confidence level of 95 %.




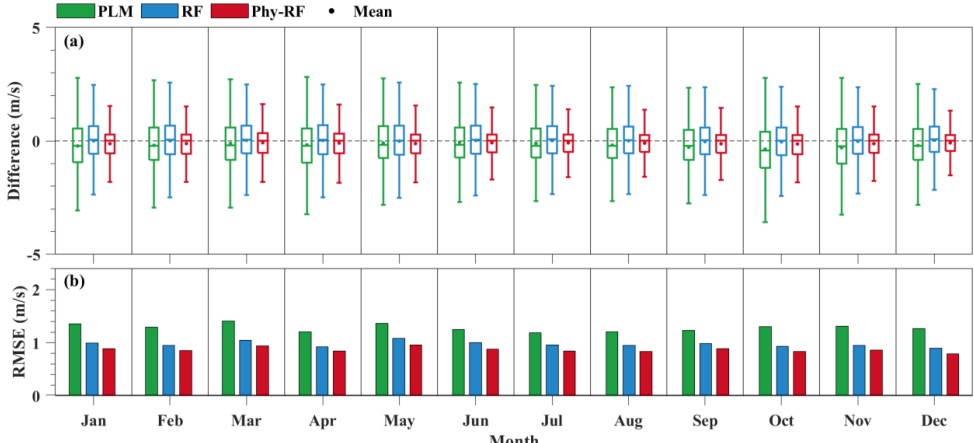

**Figure 8**. Annual cycle of the difference (a) and RMSE (b) between the estimated $WS_{100}$ and observed $WS_{100}$ for the PLM, RF and Phy-RF models. The green, blue and red colors represent the PLM, RF and Phy-RF methods, respectively.



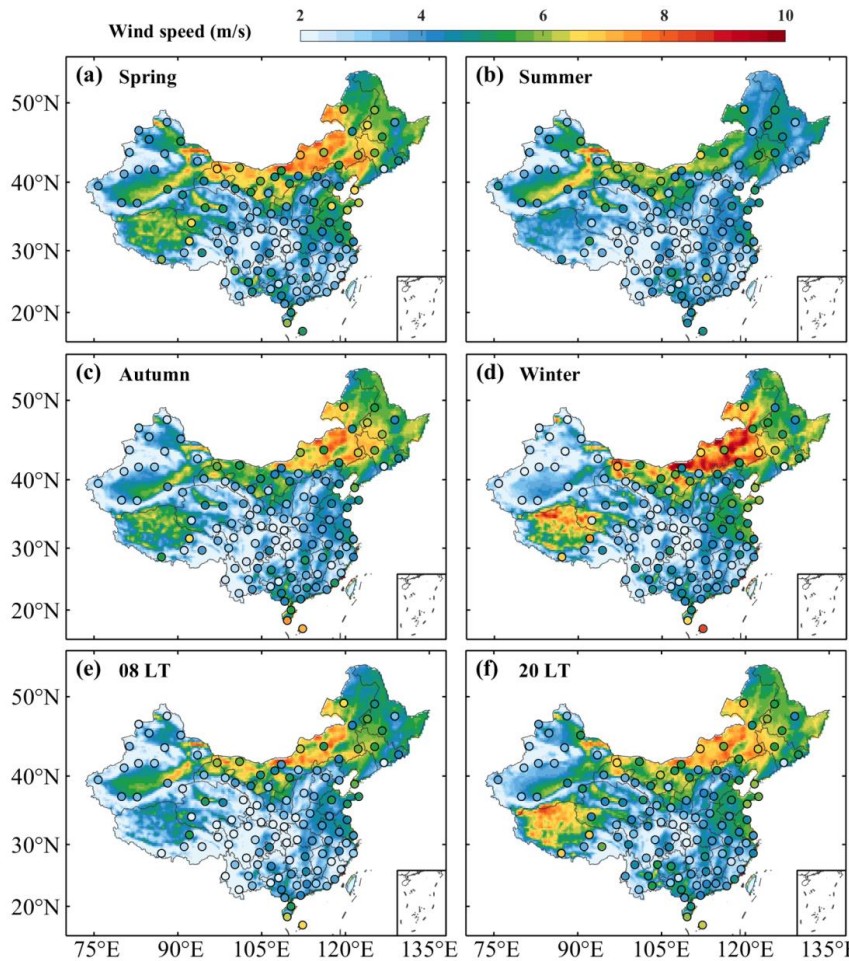

**Figure 9.** Spatial distributions of the mean $WS_{100}$ from ERA5 (color shaded) and Phy-RF model (color dots) in (a) spring, (b) summer, (c) autumn, (d) winter, (e) 0800 LT, and (f) 2000 LT.



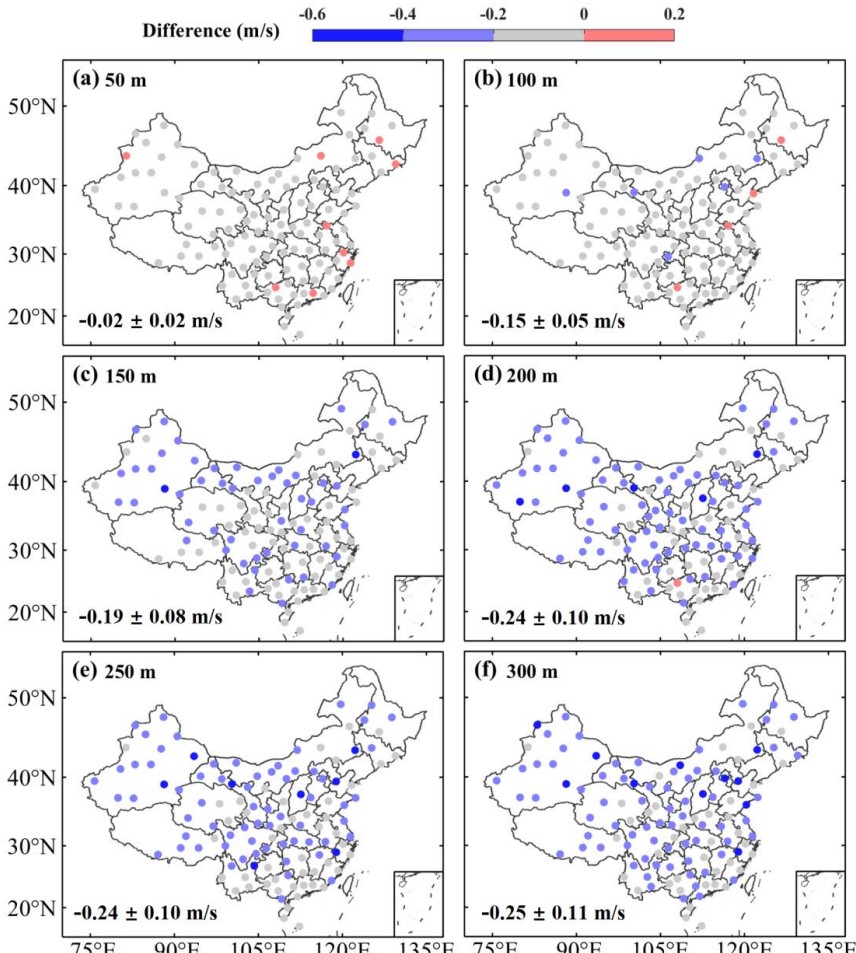

**Figure 10**. The spatial distributions of the difference between the estimated wind speed and observed wind speed for the Phy-RF model over 120 radiosonde stations in China at different heights: (a) 50 m, (b) 100 m, (c) 150 m, (d) 200 m, (e) 250 m, (f) 300 m.





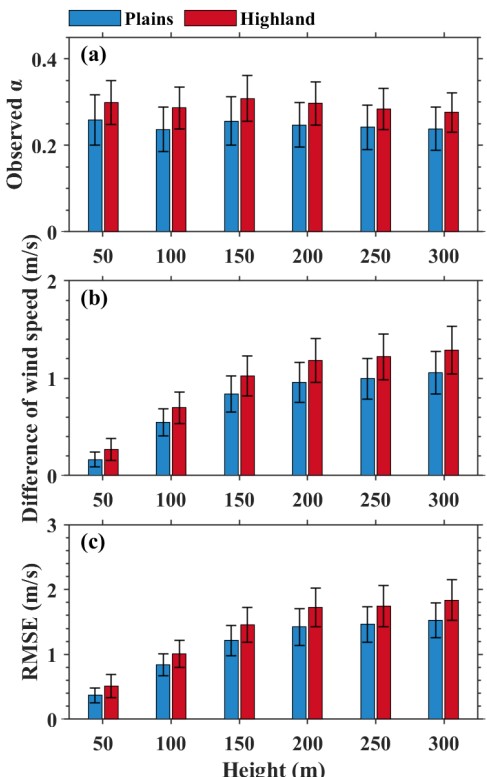

**Figure 11**. (a) The mean $\alpha$ observed from RS as a function of height. The (b) difference and (c) RMSE between the estimated wind speed and observed wind speed as a function of height. Blue and red boxes represent the results under plains and highland areas, respectively.





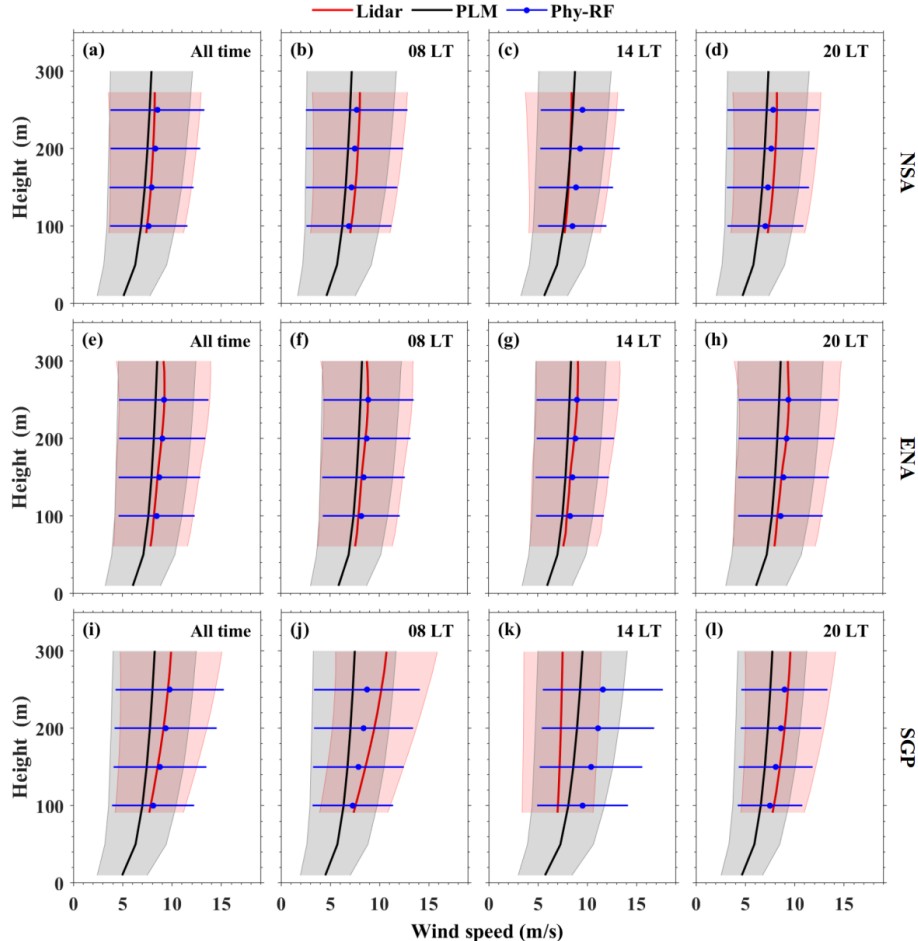

**Figure 12**. Vertical profiles of the wind speed from different methods at three ARM sites: (a-d) NSA, (e-h) ENA, and (i-l) SGP. Red, black, and blue lines represent mean wind profile from Lidar, the PLM, and the Phy-RF, respectively, and their corresponding color shading areas represent the standard deviation.

570