# Peer review of "Extending wind profile beyond the surface layer by combining physical and machine learning approaches"

_EGUsphere, 2023_

## Referee Comment (RC2)

**Review of egusphere-2023-2727: "Extending wind profile beyond the surface layer by combining physical and machine learning approaches" by Liu et al.**

**General comment**

This study introduces a Phy-RF method to extend wind profiles beyond the surface layer, overcoming limitations of the traditional model based on the Monin–Obukhov similarity theory. By combining the power law model (PLM) with the random forest (RF) algorithm, the Phy-RF method addresses errors in the PLM above the surface layer attributed to the α setting. Comparing performance over China, the Phy-RF model outperforms PLM and RF, demonstrating better accuracy and stability. Temporally, it is not significantly affected by seasonal variations but shows limitations during specific time periods. Spatially, the model performs worse in highland areas due to the absence of consideration for terrain factors.

After some minor revisions, I am in favor, that this paper gets published in ACP. The text in general should be carefully checked during the English language copy-editing process.

**Specific comments**

1. The acronym Phy-RF confuses me a bit, because you state in line 11-12: "…we propose a novel method that combines the power law method (PLM) with the random forest (RF) algorithm to extend wind profiles beyond the surface layer, called the Phy-RF method."

   Why you do not use PLM-RF as acronym if it is based on PLM. RF-PHY acronym (Radio Frequency Physical Layer) is also used in the wireless communication sector, and you may want to separate the name of your new method better.

2. In the introduction, I find the absence of a concise overview and reference of the Prandtl-layer, which encompasses the initial tens of meters within the atmospheric boundary layer.

3. The Root Mean Squared Error (RMSE) quantifies the accuracy of a regression model in predicting the response variable's value in absolute terms, whereas R-Squared measures how effectively the predictor variables account for the variability in the response variable.

   I encourage the authors to also have a look and include R-Squared or Adjusted R-Squared metrics in their model evaluation. If the authors stick with RMSE, I think they must better justify their decision.

4. Just a curiosity, now you focused on different land-cover types, can you make a statement about the performance of the Phy-RF model above water surfaces yet?

   The emphasis was on comparing the performance over China. Do you plan to investigate a more global performance estimate of the models in the future?

**Technical corrections**

1. Lines 25-26: "These findings have great implications for the weather, climate and renewable energy." The noun of the sentence is missing. Maybe add in the end of the sentence the word "sector" or "research."

2. Lines 37-38: "…, in which can be assimilate into atmospheric models to produce global wind profile products." This sounds a bit wrong. I suggest writing: "Satellite observations, such as those from Aeolus, can provide horizontal line-of-sight wind profile data that can be assimilated into atmospheric models to generate global wind profile products."

3. Line 41: I suggest to better formulate the following part: "…ground-based observations like wind tower, wind profile radar, and wind profile lidar…" into "…ground-based wind measurements from towers, radar or lidar-based profilers…".

4. Line 90ff: I recommend not to describe the color bar in the text here. Just refer to the Figure 1 and maybe specify the land cover types in the caption of this Figure, if needed.

5. Line 157: Here you talk about previous studies, but no references are given. Either add the references again or reformulate.

6. Delete full stop in title of Section 4.1.

7. Wording in title of Section 4.2 wrong. I guess should be "Wind speed evaluation of the Phy-RF model".

8. Paragraph 346-349: First sentence misses something g in the end like e.g. "sector": "…implications for the weather, climate and renewable energy sector.". Please also reformulate the second part. What is meant by "limitations of data time"? This is not clear to me.

9. Figure 3: I suggest adding the units in the y-axis's captions. The scatter points, for my impression overlap to strong here and this could lead to misinterpretations by the reader.

10. Figure 4: The figure does look too pixelated, please increase the resolution of this figure. In 4(c) correct "Wooldland" to "Woodland". The typo ("wooldland") is also in the caption of the figure.

11. Figure 5: Here please write in the caption the meaning of all variable acronyms.

---

## Author Comment (AC1)

**Response to Reviewer #1's Comments**

The paper introduces the Phy-RF model, an innovative approach combining physical wind profile models with machine learning to extend wind profile estimations beyond the surface layer. Through comprehensive analysis and validation, the study demonstrates that the Phy-RF model offers a more accurate estimation of wind speeds at 100 meters, outperforming both the traditional power law method (PLM) and random forest (RF) machine learning algorithm alone. In general, this paper is well written. I recommend the publication of this manuscript, while I have some comments below for the authors to address.

***Response: We greatly appreciated the reviewer's comments on our manuscript, which greatly improve the quality of our manuscript. We have made efforts to adequately address the reviewers' concern one by one. For clarity purpose, here we have listed the reviewer' comments in plain font, followed by our response in bold italics.***

1. The paper compares the seasonal mean of ERA-5 and Phy-RF. It could also benefit from a direct comparison between the Phy-RF model outputs and ERA-5 wind speed data at 100 meters using scatter plots. Since the model uses ERA-5 as the input, such a comparison would be beneficial for evaluating the Phy-RF model's performance.

***Response: Per your kind suggestion, we made a direct comparison between the PLM-RF model outputs and ERA-5 wind speed data at 100 meters using scatter plots, as shown in the Fig. S6. In addition, the acronym "Phy-RF" is modified to "PLM-RF" based on the reviewer #2's comment.***

***As a result, the following sentences has been added to the end of the first paragraph of Section 4.2 "Wind speed evaluation of the PLM-RF model" in the revised manuscript:***

***"In addition, the comparisons between the WS100 from ERA5 and from PLM-RF model for different periods are shown in Fig. S6. Although the output of the PLM-RF model has a good correlation with the $WS_{100}$ from ERA5, there exist still some differences. Most of the $WS_{100}$ from the PLM-RF model are greater than that of***

*ERA5 when the wind speed is high. This is because the Δα is introduced in the PLM-RF model, which makes the model tend to produce large output values."*

[Figure]

***Fig. S6. Comparisons between the WS₁₀₀ from ERA5 and from PLM-RF model in (a) spring, (b) summer, (c) autumn, (d) winter, (e) 0800 LT, and (f) 2000 LT.***

2. While the study incorporates data from several ARM sites, there appears to be insufficient evaluation using these datasets. Since averaged profiles may not capture the full extent of discrepancies between the Phy-RF model estimates and observations, relying on mean profiles from lidar cannot accurately represent accuracy.

***Response: Good points! To better reflect the performance of Phy-RF model, we investigated the diurnal variations of $R^2$, MAE and RMSE between the WS₁₀₀ calculated by PLM-RF model and the WS₁₀₀ observed by Doppler wind lidars at three ARM sites, as shown in Figure 13.***

*Accordingly, the accompanied descriptions have been supplemented in the revised manuscript. "To further evaluate the performance of the PLM-RF model, the diurnal variations of $R^2$, MAE and RMSE between the $WS_{100}$ calculated by PLM-RF model and the $WS_{100}$ observed by Lidar are investigated in Figure 13."*

[Figure]

*Figure 13. Diurnal variations of $R^2$, MAE and RMSE between the $WS_{100}$ calculated by PLM-RF model and the $WS_{100}$ observed by Lidar at (a) NSA, (b) ENA, and (c) SGP sites. The black, blue and red lines represent the $R^2$, MAE and RMSE, respectively.*

3. More detailed statistical analysis, such as examining the mean absolute error (MAE), could enhance our understanding of the model's performance. I suggest including the MAE to evaluate the differences between Phy-RF, ERA-5, and Lidar.

*Response: Good suggestion! In the revised manuscript, the statistical parameters such as coefficient of determination ($R^2$), mean absolute error (MAE) and root mean squared error (RMSE) are used to evaluate the differences between PLM-RF, ERA-5, and Lidar. The modifications can be seen in Fig.7, Fig.8, Fig.13 and Fig. S6-S8.*

4. I noticed there are notable discrepancies in the mean profiles of Phy-RF and Lidar over the SGP. What factors lead to such biases? Meanwhile, Are such biases also included in the ERA-5?

*Response: From the point of our view, the factors that may lead to such noticeable biases (at the very least) are as follows:*

*(1) The generalization of the PLM-RF algorithm depends on the training and test samples. However, the training and test samples of the PLM-RF model were obtained from soundings at 0800 and 2000 LT, which do not actually contain any in situ measurements from the period 1100 to 1500 LT. This means that the PLM-RF model has no generalization at noon, resulting in poor accuracy of the PLM-RF model during 1100 to 1500 LT.*

*(2) The SGP site are located over land, with significant diurnal variations in wind speed compared to NSA and ENA sties. Due to the lack of observational constraints from 1100 to 1500 LT, the low performance of the PLM-RF model is evident during the daytime at SGP sites.*

*These two factors lead to the noticeable discrepancies (at 1400 LT) in the mean profiles of Phy-RF and Lidar over the SGP site. Due to the biases caused by the model itself, it also occurred in the comparison with ERA-5.*

*Therefore, we added one sentence in the conclusions part: "Therefore, it is not recommended to use the PLM-RF model for the time period 1100 to 1500 LT over highland areas before including observation data to constrain the model."*

5. The paper may include its discussion on the limitations of the Phy-RF model, particularly how it performs under extreme events.

*Response: Per your kind suggestion, we discussed the limitations of the PLM-RF model in the conclusions part of this revised manuscript, which is shown as follows:*

*"The limitation of the PLM-RF model is that the performance of PLM-RF model is affected by diurnal variation and terrain. The generalization of the RF model depends on whether the training samples contain sufficient sample inputs. The training samples of the PLM-RF model do not contain in situ measurements from*

*the time period 1100 to 1500 LT, resulting in relatively poor accuracy during this period. Similarly, the RMSE of the wind profiles is relatively larger at highland areas, which is likely due to the fact that the influence of terrain was not considered in the construction of the PLM-RF model. Therefore, it is not recommended to use the PLM-RF model for the period from 1100 to 1500 LT over highland areas before adding observation data to retrain the model."*

*In addition, the Fig. 7 shows that the PLM-RF model has better accuracy and stability compared to PLM and RF. Especially under high wind speed events, the output of PLM is significantly low, while the PLM-RF model can effectively correct this underestimation. The modifications in the revised manuscript are as follows:*

*"Overall, the advantage of the PLM-RF model is that it can provide more accurate wind profiles than the PLM, especially when the actual wind speed is high."*

6. Despite a worse performance compared to Phy-RF, the traditional PLM also seems good (Figure 7). It would be beneficial to discuss more clearly the potential applications and advantages of the Phy-RF approach, particularly in scenarios where the PLM and ERA-5 may fall short.

*Response: Points have been well taken. Per your suggestion, we discussed the potential applications and advantages of the PLM-RF model in the conclusions part, which is shown as follows:*

*"Overall, the advantage of the PLM-RF model is that it can provide more accurate wind profiles than the PLM, especially when the actual wind speed is high. Moreover, the PLM-RF model is not affected by seasonal variation. This is because the RF model is data driven. The training sample of the PLM-RF model contains enough samples from four seasons. The PLM-RF model is recommended for areas with high wind speeds, such as coastal areas."*

---

## Author Comment (AC2)

**Response to Reviewer #2's Comments**

This study introduces a Phy-RF method to extend wind profiles beyond the surface layer, overcoming limitations of the traditional model based on the Monin–Obukhov similarity theory. By combining the power law model (PLM) with the random forest (RF) algorithm, the Phy-RF method addresses errors in the PLM above the surface layer attributed to the α setting. Comparing performance over China, the Phy-RF model outperforms PLM and RF, demonstrating better accuracy and stability. Temporally, it is not significantly affected by seasonal variations but shows limitations during specific time periods. Spatially, the model performs worse in highland areas due to the absence of consideration for terrain factors. After some minor revisions, I am in favor, that this paper gets published in ACP.

*Response: We greatly appreciated the reviewer's comments on our manuscript, which greatly improve the quality of our manuscript. We have made efforts to adequately address the reviewers' concern one by one. For clarity purpose, here we have listed the reviewer' comments in plain font, followed by our response in bold italics.*

1. The text in general should be carefully checked during the English language copy-editing process.

*Response: Thanks for pointing these issues out. We tried our best to correct spelling and grammatical errors in the revised manuscript.*

2. The acronym Phy-RF confuses me a bit, because you state in line 11-12: "…we propose a novel method that combines the power law method (PLM) with the random forest (RF) algorithm to extend wind profiles beyond the surface layer, called the Phy-RF method." Why you do not use PLM-RF as acronym if it is based on PLM. RF-PHY acronym (Radio Frequency Physical Layer) is also used in the wireless communication sector, and you may want to separate the name of your new method better.

*Response: Good point! To avoid misunderstandings, the proposed method of ""Phy-RF" has been revised to "PLM-RF" throughout this whole revised manuscript, from the main text to figures.*

3. In the introduction, I find the absence of a concise overview and reference of the Prandtl layer, which encompasses the initial tens of meters within the atmospheric boundary layer.

*Response: Agreed! It is well known that Dr. Prandtl proposed the concept of Prandtl layer in 1904. In the thin layer near the solid wall, the influence of viscous force cannot be ignored, and this thin layer is called the Prandtl boundary layer. Therefore, we added a concise overview and reference of the Prandtl layer in the introduction, which is shown as follows:*

*"The Prandtl layer encompasses the initial tens of meters within the atmospheric boundary layer (Anderson, 2005)."*

*Anderson, J. D.: Ludwig Prandtl's boundary layer. Physics today, 58(12), 42-48, https://doi.org/10.1063/1.2169443, 2005.*

4. The Root Mean Squared Error (RMSE) quantifies the accuracy of a regression model in predicting the response variable's value in absolute terms, whereas R-Squared measures how effectively the predictor variables account for the variability in the response variable. I encourage the authors to also have a look and include R-Squared or Adjusted R-Squared metrics in their model evaluation. If the authors stick with RMSE, I think they must better justify their decision.

*Response: According to your suggestion, the statistical parameters such as coefficient of determination ($R^2$), mean absolute error (MAE) and root mean squared error (RMSE) are used in the model evaluation. The modifications can be seen in Fig.7, Fig.8, Fig.13 and Fig. S6-S8.*

5. Just a curiosity, now you focused on different land-cover types, can you make a statement about the performance of the Phy-RF model above water surfaces yet? The emphasis was on comparing the performance over China. Do you plan to investigate a more global performance estimate of the models in the future?

*Response: Because the radiosonde stations are mainly located over land and the drifting route of sounding balloon varies sharply over time and space, we are unable to analyze the performance of the PLM-RF model on the water surface. In the future,*

*we plan to use global RS observation data to train and test the PLM-RF model, and evaluate its performance on a global scale.*

*According to your suggestion, we make a statement about the performance of the PLM-RF model above water surfaces in the last paragraph of the conclusions. The modifications in the revised manuscript are as follows:*

*"However, due to the limitations in data size and terrain factors, the performance of the PLM-RF model above water surfaces is uncertain. In the future, the global RS observation data will be used to train and test the PLM-RF model, and evaluate its performance on a global scale."*

6. Lines 25-26: "These findings have great implications for the weather, climate and renewable energy." The noun of the sentence is missing. Maybe add in the end of the sentence the word "sector" or "research."

*Response: Good suggestion! We added the "sector" in the end of the sentence.*

7. Lines 37-38: "…, in which can be assimilate into atmospheric models to produce global wind profile products." This sounds a bit wrong. I suggest writing: "Satellite observations, such as those from Aeolus, can provide horizontal line-of-sight wind profile data that can be assimilated into atmospheric models to generate global wind profile products."

*Response: Amended as suggested.*

8. Line 41: I suggest to better formulate the following part: "…ground-based observations like wind tower, wind profile radar, and wind profile lidar…" into "…ground-based wind measurements from towers, radar or lidar-based profilers…".

*Response: Amended as suggested.*

9. Line 90: I recommend not to describe the color bar in the text here. Just refer to the Figure 1 and maybe specify the land cover types in the caption of this Figure, if needed.

*Response: Good suggestion! We deleted the description about color bar in here.*

10. Line 157: Here you talk about previous studies, but no references are given. Either add the references again or reformulate.

*Response: We added the references.*

11. Delete full stop in title of Section 4.1.

*Response: Amended as suggested.*

12. Wording in title of Section 4.2 wrong. I guess should be "Wind speed evaluation of the PhyRF model".

*Response: The title of Section 4.2 modified to "Wind speed evaluation of the PLM-RF model".*

13. Paragraph 346-349: First sentence misses something g in the end like e.g. "sector": "…implications for the weather, climate and renewable energy sector.".

*Response: We added the "sector" in the end of the sentence.*

14. Please also reformulate the second part. What is meant by "limitations of data time"? This is not clear to me.

*Response: Amended as "However, due to the limitations in data size and terrain factors, the performance of the PLM-RF model above water surfaces is uncertain. In the future, the global RS observation data will be used to train and test the PLM-RF model, and evaluate its performance on a global scale."*

15. Figure 3: I suggest adding the units in the y-axis's captions. The scatter points, for my impression overlap to strong here and this could lead to misinterpretations by the reader.

*Response: Amended as suggested.*

16. Figure 4: The figure does look too pixelated, please increase the resolution of this figure. In 4(c) correct "Wooldland" to "Woodland". The typo ("wooldland") is also in the caption of the figure.

*Response: Amended as suggested.*

17. Figure 5: Here please write in the caption the meaning of all variable acronyms.

*Response: Amended as suggested.*